# Complex Regional Pain Syndrome after Distal Radius Fracture—Case Report and Mini Literature Review

**DOI:** 10.3390/jcm13041122

**Published:** 2024-02-16

**Authors:** Michał Świta, Paweł Szymonek, Konrad Talarek, Agnieszka Tomczyk-Warunek, Karolina Turżańska, Agnieszka Posturzyńska, Anna Winiarska-Mieczan

**Affiliations:** 1Department of Rehabilitation and Orthopedics, Medical University of Lublin, Jaczewskiego 8, 20-954 Lublin, Poland; michal.swita10@gmail.com (M.Ś.); szymonek.p.lbl@gmail.com (P.S.); konradtalarek18@gmail.com (K.T.); agnieszka.posturzynska@umlub.pl (A.P.); 2Laboratory of Locomotor Systems Research, Department of Rehabilitation and Physiotherapy, Medical University of Lublin, Jaczewskiego 8, 20-954 Lublin, Poland; 3Department of Bromatology and Nutrition Physiology, Institute of Animal Nutrition and Bromatology, University of Life Sciences in Lublin, Akademicka St. 13, 20-950 Lublin, Poland; anna.mieczan@up.lublin.pl

**Keywords:** complex regional pain syndrome (CRPS), distal radius fracture (DRF), mirror therapy

## Abstract

This study explores the impact of the complex regional pain syndrome (CRPS) on the lives and mobility of patients, with a particular focus on its emergence as a late complication of distal radius fractures (DRFs), a common occurrence, especially among an aging population. The absence of a standardized treatment for the CRPS and the challenge of predicting its occurrence make it a complex medical issue. This research aims to shed light on the effects of treating the CRPS through a case study involving a 75-year-old woman with untreated osteoporosis who experienced a Colles fracture after a fall. The initial management involved repositioning and stabilizing the fractured forearm with a plaster cast, followed by an operation using percutaneous pinning via a Kirschner wire. Subsequently, the patient developed CRPS symptoms and was admitted to the rehabilitation department three months post-fracture. The affected forearm exhibited swelling, warmth, pain, and severely limited range of motion. Treatment involved a combination of medications, physiotherapy, and kinesiotherapy. Significantly, the patient experienced notable improvement following these interventions. This study underscores the absence of a definitive standard for CRPS treatment but suggests that proper rehabilitation and pharmaceutical interventions can contribute positively to patient outcomes. The case further highlights the potential association between DRF and CRPS development, emphasizing the need for continued research in this field.

## 1. Introduction

### 1.1. Distal Radius Fractures: Presentation and Epidemiology

Distal radius fractures (DRFs) are among the most common fractures, and their incidence is constantly increasing. The groups most likely to suffer from these fractures are adults over 50 and children under 18 [1]. In people over 65, only hip fractures are more common than distal radius fractures. The highest frequency of DRF occurs in the age group of 75–84 years (351.5 per 100,000 annually) [2]. The frequency of distal radius fractures in particular sexes changes depending on age. In groups up to 50, this fracture is more often observed in men. After the age of 50, such a fracture predominates among women. Vehicle accidents and sports activities contribute the most to fractures in the young group. In the second group, the main reason is low-energy trauma [1,3,4].

### 1.2. Distal Radius Fractures: Treatment and Complications

Among the therapeutic approaches, we can delineate non-operative methods, encompassing closed reduction and the application of a plaster cast. Alternatively, surgical interventions are available, encompassing open reduction and internal fixation (ORIF), Kirschner wire stabilization, and external fixation [5]. The prevalence of complications following non-surgical interventions is the primary driver for the increased utilization of surgical methods [6,7,8,9]. Distal radius fractures or their management may give rise to complications, categorized into three groups. Immediate complications, constituting the first group, involve nerve injury, skin damage during manipulation, and compartment syndrome. The second group encompasses early complications (within six weeks), such as loss of reduction, infection, and tendon rupture. Late complications, occurring beyond six weeks, form the last group, and include arthrosis and the complex regional pain syndrome. 

#### 1.2.1. Complex Regional Pain Syndrome: Natural History and Presentation

The complex regional pain syndrome (CRPS) is characterized by persistent pain, predominantly impacting the limbs [10,11,12]. The condition is categorized into two types, with Type I manifesting when symptoms arise subsequent to a traumatic event [13]. CRPS-1 frequently occurs in cases of upper limb injuries stemming from fractures and surgical procedures, with distal radius fractures (DRFs) being a common cause [14]. Type II, on the other hand, is linked to nerve injuries [13]. A separate source on the complex regional pain syndrome suggests that both types can result from trauma, with the distinction hinging on evident nerve injury, observed exclusively in Type II. This differentiation underlies the divergence in chronic pain types, characterized as mainly nociceptive in CRPS I and mainly neuropathic in CRPS II [15]. Some patients may only meet partial criteria for CRPS I or II, leading to the possibility of a third type diagnosis in the absence of a better explanation for the symptoms and signs [12,16]. The pathophysiology of the complex regional pain syndrome arises from an inadequate body response to tissue injury and exhibits a multifactorial nature. Neurogenic inflammation, vessel dysfunction, heightened sensitivity of nociceptive receptors, and impaired nerve plasticity collectively influence the clinical manifestations of the CRPS. Each patient experiences unique contributions from these four factors, giving rise to various clinical presentations of this disease [17,18]. Symptoms typically emerge rapidly, within days or weeks, predominantly affecting one limb and localizing to its distal part. The condition is characterized as acute (lasting less than two months) or chronic (persisting beyond two months). For over 80% of patients, symptoms resolve within 18 months, while persistent symptoms beyond this period often lead to irreversible changes [19].

#### 1.2.2. Complex Regional Pain Syndrome: Diagnosis and Epidemiology

The Budapest Criteria are used to diagnose clinical CRPS [20]. They were developed in 2003 and have enhanced specificity compared to the old IASP (International Association for the Study of Pain) criteria [13,21]. The CRPS is diagnosed when alternative diseases are eliminated [20]. The Budapest criteria organized a list of signs and symptoms into four categories. The diagnosis of the CRPS is made when the patient reports at least one symptom in at least three categories or when the physician detects at least one symptom in at least two categories (the list of categories of specific signs and symptoms is given in the table below). In addition, the CRPS can be diagnosed if a patient reports continuous pain, which is disproportionate to any inciting event, and in cases when no other diagnosis can better explain the signs and symptoms [22] (Table 1). Besides the Budapest criteria, bone scans are also considered a valuable CRPS diagnostic method [23].

Significant risk factors for CRPS-1 include female sex, rheumatoid arthritis (RF+), assisting ulnar fractures, fibromyalgia, external fixation and open fractures. Furthermore, RA has the highest odds ratio, followed by female sex and assisted ulnar fracture. Age and psychiatric factors are irrelevant [14,24].

The epidemiology of CRPS is too complex to estimate due to the small amount of data on it. In a retrospective study where data were gathered from 600,000 patients, the incidence rate was 26.2 per 100,000 people. Incidences are higher in females, most often in the 61–70 age range. The disease more frequently affects the upper limb. In total, 44% of subjects suffer a fracture before the symptoms of the CRPS appear [25].

#### 1.2.3. Complex Regional Pain Syndrome: Phases of CRPS

The development of the CRPS is divided into three phases. The first one is called the acute phase and lasts about three months. There is pain, edema, warming, and redness. In addition, increased sweating and increased growth of hair and nails can occur.

The second one is the dystrophic phase (3–9 months). Pain with cold and sweaty skin, loss of hair, fragility of nails, osteoporosis, and lividness are symptoms of this phase.

The third one is called the atrophic phase (>9 months) due to the atrophy of the skin and muscles. The pain is spread proximally to the affected area. Contractures of joints can occur. In radiological examinations, advanced bone lesions may appear [26].

#### 1.2.4. Complex Regional Pain Syndrome: Prevention

Proper post-traumatic treatment from the first hours after injury or surgery is crucial. Appropriate pain relief (immobilization) and anti-swelling measures should be taken, as well as avoiding the inhibition of blood outflow. The patient should be instructed about the need for constant elevation of the affected limb and weight-bearing of the limb as quickly as reasonably possible. Early implementation of active exercises in joints that are not immobilized is also critical [27]. It has also been proven that daily vitamin C intake (500–1000 mg) reduces the risk of developing the CRPS after orthopedic procedures and injuries [28,29].

#### 1.2.5. Complex Regional Pain Syndrome: Treatment

The risk of developing the CRPS depends on the actions taken after the injury. The limb should be immobilized, and appropriate elevation is also necessary. It is crucial to implement exercises and load-bearing the limb as early as possible [30].

Restoring the function of an affected limb is a central principle in the treatment of the CRPS, so pain management is considered secondary to function, even though it is crucial as well [30]. There is no single proper method of treatment due to the multifactorial nature of the CRPS. The treatment focuses on improving the patient’s functioning and reducing pain. It relies on a comprehensive approach that includes psychological, medical, physical, and occupational therapy components [31].

Therapy consists of pharmacological treatment and rehabilitation. In pharmacotherapy, non-steroidal anti-inflammatory drugs, glucocorticosteroids, calcitonin, bisphosphonates, sympatholytic drugs such as carbamazepine and gabapentin, antidepressants (amitriptyline), and supplementation of vitamin D are used. Stellate ganglion blocks and lumbar sympathetic blocks are also reported as effective methods of CRPS pain treatment [32]. Rehabilitation in the initial phase focuses on physical therapy treatments that are intended to reduce pain and hypersensitivity. When the pain is controlled well, kinesitherapy treatments are introduced. Due to the length of treatment, psychological support is an essential element which reduces the patient’s stress and increases his/her motivation [26].

Promising neurocognitive methods comprise prism adaptation (PA), mirror therapy, and graded motor imagery (GMI). A double-blind, randomized controlled trial (RCT) consisting of 42 patients demonstrated pain relief in those undergoing PA. This method has been successfully used in hemispatial neglect after brain injury. Another RCT consisting of 66 participants related to CRPS treatment post-DRF showed a short-term improvement in patients’ conditions [33,34]. Existing surgical methods include epidural unilateral stimulation or peripheral nerve stimulation [35,36]. Cases have been described in the literature where patients undergoing extended, ineffective drug therapy after the procedure returned to full health [35].

Recent years have seen many innovative concepts in CRPS treatment; however, there is no “golden standard” yet.

## 2. Case Report

### 2.1. Patient

The focus of this investigation is a 75-year-old female with the complex regional pain syndrome (CRPS) following a distal radius fracture (DRF). The patient received treatment at the Department of Rehabilitation and Orthopedics. The incident leading to the fracture occurred three months prior to her admission to the rehabilitation department in December 2022, involving a fall from her a standing position. The patient’s medical history includes mild hypertension, hyperthyroidism, and untreated osteoporosis, as evidenced via a DEXA scan conducted two weeks post-injury, revealing a hip T-score of −2.6 and a lumbar spine T-score of −3.4—suggesting osteoporosis as the likely cause of the fracture. The patient is currently prescribed Nebivolol 5 mg 1 × 1/4, Rosuvastatin 5 mg 1 × 1, and Acetylsalicylic Acid 1 × 1, the latter of which was discontinued following the fall.

### 2.2. Fracture and Initial Treatment

The patient came to the emergency department with intense pain, loss of function, and swelling in the left forearm following a fall from her a standing position onto a hard surface. A physical examination and X-ray of the left upper limb led to the diagnosis of a Colles fracture. The limb was realigned and secured in a plaster cast. After 28 days, surgery involving Kirschner wire stabilization of the radial bone was performed (Figure 1). Following hospital discharge, the patient underwent five outpatient follow-up visits, which included X-rays of the left upper limb and changes to the plaster cast. Notably, symptoms indicative of developing the complex regional pain syndrome (CRPS) remained unnoticed until the last control visit. During the seventh week post-surgery, an attempt to remove the Kirschner wires was made. However, due to the observation of troubling symptoms, the patient was subsequently referred to the Department of Rehabilitation and Orthopedics in March 2023.

### 2.3. After Admission to the Rehabilitation Department

#### 2.3.1. Patient Examination

Following admission to the Department of Rehabilitation and Orthopaedics, radiological examination revealed complete fusion of the fracture. However, bone tissue rarefaction in the left upper limb was evident, manifesting as spots affecting the left carpal bones, left radial bone, and the proximal section of the I-V left metacarpal bones (refer to Figure 2). The physical examination of the affected limb indicated severe pain (rated 8/9 on the VAS scale), hyperalgesia, swelling, tightness of the skin, and a glowing appearance The functional capacity of the patient’s hand was significantly impaired, with active movement limited to slight extension of the wrist joint (20°) due to intense pain and edema. The patient was unable to form a fist, and passive movements were also constrained. Specifically, flexion of the left wrist joint, radial and ulnar adduction were 0°, while extension of the wrist joint was partially limited to 60°. Although extension of metacarpophalangeal (MCP) joints, proximal interphalangeal (PIP) joints, and distal interphalangeal (DIP) joints were normal, they were accompanied by severe pain. MCP Joint flexion, abduction, and adduction were limited to 0°, while flexion of the PIP and DIP joints was restricted to 5–10°. Passive supination and pronation were also painfully limited to around 20°. High stiffness in the wrist, MCP, PIP, and DIP joints was observed. Despite significant swelling, no muscle atrophy was noted on the palmar and dorsal surfaces of the left hand or the forearm. The patient scored 0 points in the Frenchay Arm Test (FAT). The remaining joints of the left limb maintained an unaffected range of motion. A comprehensive blood test indicated specific parameter levels: PTH (parathormone): 29.60 pg/mL; calcium: 9.10 mg/dL; 25(OH)D3: 39.4 ng/mL; b-ALP (bone alkaline phosphatase): 4.9 (normal range: 5.3–24.6); CRP (C-reactive protein) <5.0. Based on the Budapest criteria, the patient exhibited at least one symptom from each category (hyperalgesia, skin temperature and color asymmetry, edema, decreased range of motion), leading to the diagnosis of the complex regional pain syndrome (CRPS).

#### 2.3.2. Pharmacology and Rehabilitation Treatment

The patient underwent a 30-day stay in the rehabilitation department, during which additional medications were prescribed alongside those previously taken. The administered medications included Vitamin D3 at a dosage of 20,000 IU once per week, Milgamma (a combination of vitamin B1, B6, and B12) at 100 mg twice a day, and Osteogenon (an ossein microcrystalline hydroxyapatite compound) at 830 mg twice a day. Pain management involved Naproxen 250 mg and Ibuprofen 200 mg, administered upon request without surpassing the specified maximum doses, coupled with a proton pump inhibitor (PPI, 20 mg given once daily) for protection. Additionally, intravenous Ibandronic acid 150 mg was administered during the first week of hospitalization (refer to Figure 3). Complementary to pharmacological interventions, the core of the treatment regimen comprised physiotherapy and kinesiotherapy. Physiotherapeutic methods encompassed low-frequency magnetic field therapy, cryotherapy, whirlpool baths, and iontophoresis with calcium carbonate, conducted five times a week. Kinesiotherapy emphasized individually tailored exercises incorporating scar therapy, manual therapy, manual dexterity exercises, and hand occupational therapy, which were assigned six times a week. Mirror therapy sessions were also integrated and conducted six times a week with a progressive duration of 10 to 30 min daily (see Figure 3). The patient’s affected hand after one week of treatment is shown in Figure 4.

#### 2.3.3. Follow-Up Outcome

The patient’s condition improved significantly after the used procedures. Pain and edema gradually decreased, and the range of motion of the left upper limb increased, which was observed when the patient performed complex manual activities, for example, playing the piano in the hospital chapel. The day before discharge from the hospital, the range of motion (ROM) in the patient’s hand joints was examined, and the results were compared to the healthy side. Based on the comparison of passive and active movements in the hand joints, it was noticed that the range of passive and active movements increased. The patient had a full range of flexion and extension of the MCP, PIP, and DIP joints. The movement of the left wrist joint remained marginally limited compared to the healthy hand. The flexion and extension of the left wrist joint were both 60°. Abduction and adduction in this joint were complete. Supination and pronation were normal. The patient could form a fist, and the left hand’s dexterity improved significantly. The patient achieved 5 points (maximal score) in the FAT. Pain on the VAS scale decreased to 1 point. After the end of the patient’s stay at the ward, she continued treatment in the outpatient rehabilitation unit to maintain good treatment results (Figure 5).

## 3. Discussion

In a study identifying risk factors for the complex regional pain syndrome (CRPS), an analysis involving 647 cases from the Danish Patient Compensation Association revealed that the female gender, upper limb involvement, and surgical intervention were identified as risk factors [37]. Notably, age was not identified as a risk factor; however, a systematic literature review emphasized that the risk increases after menopause, primarily due to osteoporosis [38,39]. The patient discussed in this report was a post-menopausal woman who underwent surgery on an upper limb after an injury. The studies related to CRPS treatment exhibit considerable uncertainty, with multiple reviews indicating a low certainty of evidence for various therapies [10,40,41]. The prevailing literature advocates a multidisciplinary approach encompassing pharmacological treatment, psychotherapy, physiotherapy, and occupational therapy [42]. 

Regarding pharmacotherapy, it typically involves steroids and non-steroidal anti-inflammatory drugs (NSAIDs). However, recent publications report a lack of evidence supporting the use of NSAIDs. Potential future or infrequently used medications include bisphosphonates, antioxidants, naltrexone, botulinum toxin type A, ketamine, or gabapentin [10]. In the presented case, based on the clinical presentation, physical examination, and X-ray, the chosen treatment approach involved rehabilitation procedures and pharmacotherapy. Notably intravenous bisphosphonates (Ibandronic acid 150 mg) were administered, targeting the patient’s low-energy fracture resulting from untreated osteoporosis. It is worth mentioning that a publication on the use of bisphosphonates in treating the CRPS suggests relief of the symptoms, particularly severe and moderate pain, mainly in patients with the CRPS of the upper limb [43]. Additionally, vitamin D3 supplementation at a dose of 2000 IU was introduced. A study on post-menopausal patients, after distal radius fractures, showed a significant association between vitamin D3 deficiency and the occurrence of the CRPS. Subjects with the CRPS exhibited markedly lower initial 25(OH)D3 serum levels than those without the CRPS. The anti-inflammatory effect of vitamin D may explain its effectiveness in preventing the CRPS [44]. In this case, the patient’s 25(OH)D3 serum levels were within the reference range during the treatment, as tested twice. Low-frequency magnetic fields, especially Bio-Electro-Magnetic-Energy-Regulation (BEMER), played a pivotal role in physical therapy during the treatment. BEMER is based on a low-intensity alternating magnetic field, and it is known for its effects on microcirculation. A pilot study demonstrated differences in pain reduction in the experimental group compared to the placebo BEMER group [45]. BEMER has shown effectiveness in the treatment of various conditions such as multiple sclerosis, knee osteoarthritis, and coronary heart disease [46,47,48,49]. A study by Benedetti et al. (2020) also suggested the effectiveness of BEMER in treating the CRPS, although it was a pilot study, indicating the need for further research [45].

Another increasingly used therapy for pain syndromes is mirror therapy (MT). During MT, the patient observes a mirror positioned between their limbs, concealing the affected limb with a reflection of the healthy limb. This technique, proven to enhance the range of motion (ROM) in patients after limb immobilization, involves phases such as the adaptive phase and movement synchronization of both limbs. Recent meta-analyses have shown the positive effects of mirror therapy on strength, ROM, speed, functional state, and balance in immobilized limbs [50,51,52].

The treatment administered to the patient in our study yielded favorable results, effectively alleviating reported pain and restoring manual dexterity to the wrist and fingers. Despite initial limitations in the patient’s left limb functionality, the entire treatment process in the rehabilitation department spanned approximately a month (30 days). This is noteworthy, considering that persistent therapy-resistant CRPS lasting over a year can significantly impact a patient’s life, leading to chronic pain, disability, sleep disturbances, and intimacy deprivation [53].

The treatment strategy aimed to promptly enhance the patient’s health, enabling her to regain independence in essential daily activities. The literature supports the significance of the treatment process in managing the CRPS, emphasizing the crucial role of rehabilitation prophylaxis in preventing complications following fractures. A study on patients with fractures revealed that late or delayed orthopedic rehabilitation increases the likelihood of developing the CRPS [54]. Early diagnosis coupled with a proper response may prevent severe complications and contribute to full functionality, as evidenced in this patient’s case.

Furthermore, a study assessing 975 patients with CRPS on the Quality of Life (QoL) scale demonstrated that exclusion from hobbies had a more pronounced negative impact on their lives than the pain itself [55]. A prompt and effective intervention, in this case, enabled the patient to return to activities such as driving a car and playing the piano. Given the patient’s age and associated mobility limitations, alleviating difficulties in performing basic activities and engaging in hobbies should be a crucial consideration in post-injury therapy.

## 4. Conclusions

Attention to complications following distal radius fractures (DRFs) is crucial, necessitating early rehabilitation and rehabilitation prophylaxis to minimize the risk of developing the complex regional pain syndrome (CRPS). Timely detection of CRPS symptoms can prevent severe complications and facilitate full functional recovery. Several significant risk factors for developing the CRPS, including female gender, surgical treatment of fractures, and fractures of the upper limb, highlight the importance of incorporating risk factor inquiries into the patient's post-fracture medical history. This practice aids in assessing the potential progression of the CRPS. Despite an exhaustive review of our case and the existing literature on the CRPS, a singular effective treatment method has not been identified. Notably, there is limited research in the available literature evaluating the impact of Bio-Electro-Magnetic-Energy-Regulation (BEMER) on CRPS treatment [45]. Our case report stands as the initial endeavor of its kind to document the utilization of this method in medical practice. The comprehensive nature of treatment, encompassing physical therapy, kinesiotherapy, and pharmacotherapy, is crucial. Further research in this direction is recommended to enhance our understanding and refine the treatment approaches regarding the CRPS.

## Figures and Tables

**Figure 1 jcm-13-01122-f001:**
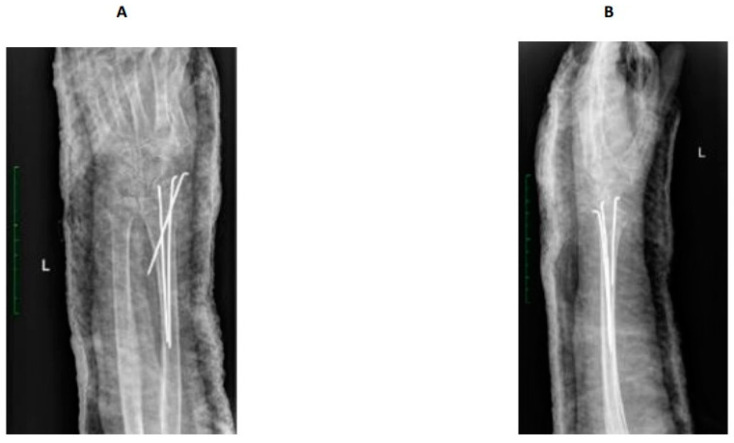
X-ray after surgery. Distal radius fracture of the left upper limb with K-wire. AP view (**A**), lateral -view (**B**).

**Figure 2 jcm-13-01122-f002:**
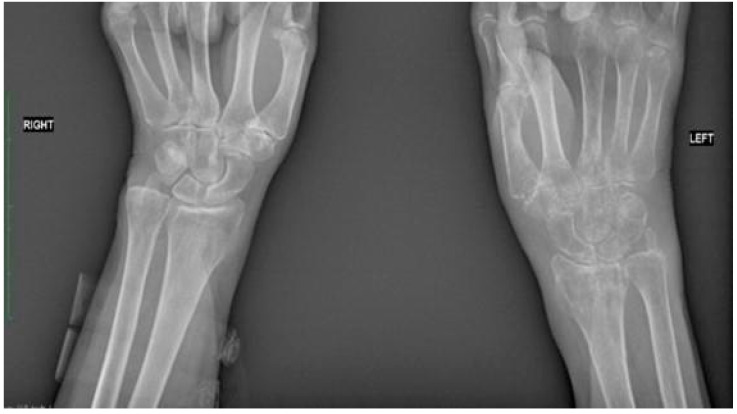
Comparative radiographic assessment of two wrists in antero-posterior plain shows no bone pathology at the right site. Bone tissue rarefaction is clearly visible on the left side and appears to be like spots. It involves the left carpal bones, the left radial bone, and the proximal part of the I-V left metacarpal bones. There is a loss of joint space between carpal bones and between a distal row of carpal and metacarpal bones.

**Figure 3 jcm-13-01122-f003:**
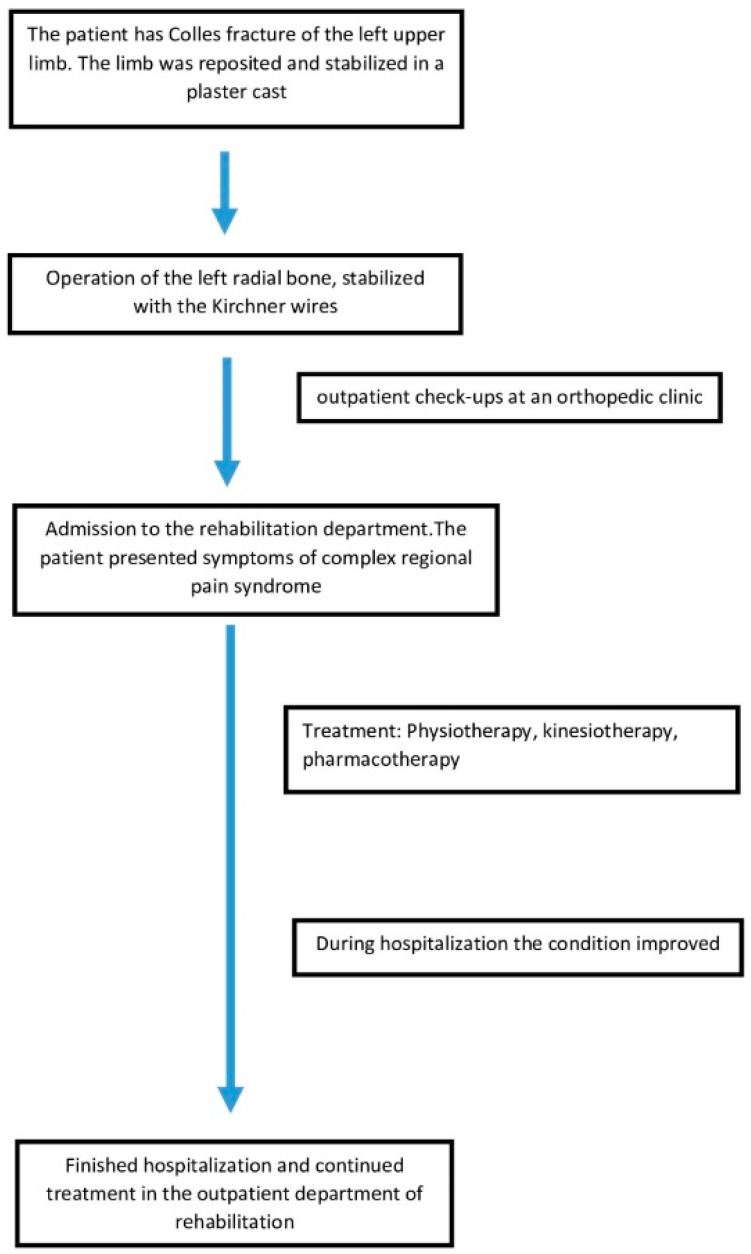
Graphical presentation of the patient’s treatment course.

**Figure 4 jcm-13-01122-f004:**
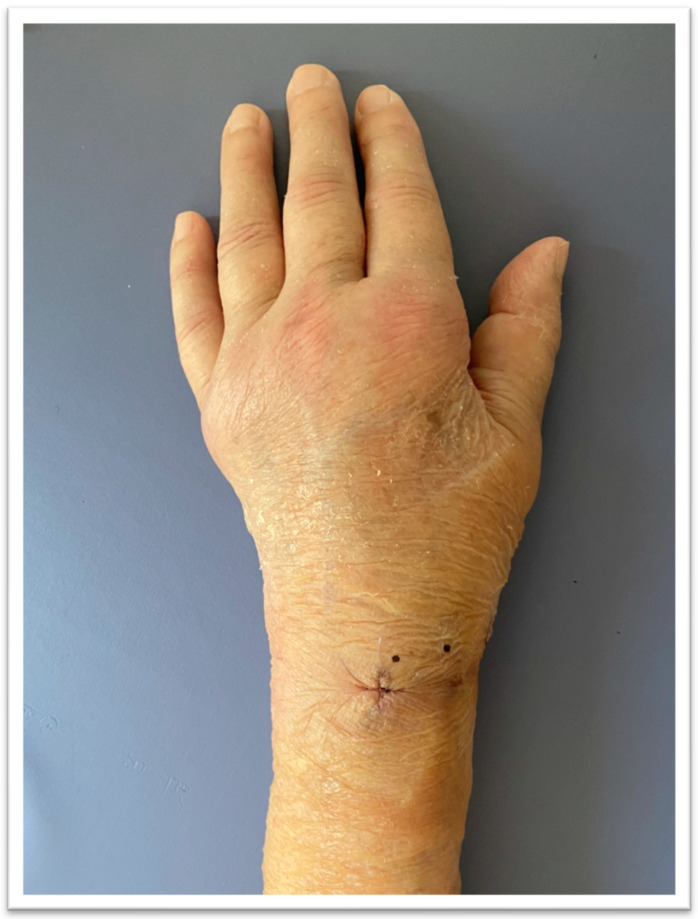
Patient’s hand one week after admission to the rehabilitation department. After one week of treatment: slight edema, decreased pain level, and fading redness. Suture after K-wire removal.

**Figure 5 jcm-13-01122-f005:**
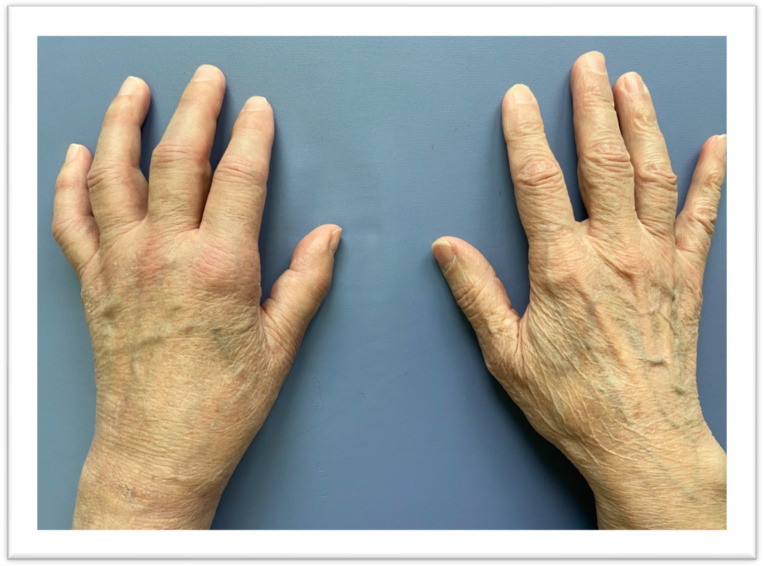
Patient’s hands after one month of treatment: improvement in function, further decrease of edema, no redness, no pain.

**Table 1 jcm-13-01122-t001:** Budapest criteria of CRPS DIAGNOSIS [21].

Categories	Objective Signs	Subjective Symptoms
1. **Sensory disorders**	hyperesthesia and/or allodynia	evidence of hyperalgesia (to pinprick) and/or allodynia (to light touch and/or deep somatic pressure and/or joint movement)
2. **Vascular disorders**	temperature asymmetry and/or skin color changes and/or skin color asymmetry	evidence of temperature asymmetry >1 °C and/or skin color changes and/or asymmetry
3. **Edema and sweating disorders**	edema or sweating changes or sweating asymmetry	evidence of edema and/or sweating changes and/or sweating asymmetry
4. **Motor/trophic disorders**	decreased range of motion and/or motor dysfunction (weakness, tremor, dystonia) and/or trophic changes (hair, nail, skin)	evidence of decreased range of motion and/or motor dysfunction (weakness, tremor, dystonia) and/or trophic changes (hair, nail, skin)

## Data Availability

Data are contained within the article.

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
