# Peer review of "Complex Regional Pain Syndrome after Distal Radius Fracture—Case Report and Mini Literature Review"

_jcm, 2024, doi:10.3390/jcm13041122_

Round 1

Reviewer 1 Report

Comments and Suggestions for Authors

I commend the authors for their hard work and contribution to the field.

1-The stellate ganglion blocks and lumbar sympathetic blocks are used for diagnosis and treatment of sympathetically mediated/maintained pain in a subset of patients diagnosed with CRPS. You could mention about stellat block in the discussion section.

2- In CRPS I, the pain is nociceptive, and in CRPS II, it is neuropathic. 

I do not agree with this sentence. Could you please add reference or rewrite the sentence.

Author Response

I commend the authors for their hard work and contribution to the field.

-Thank you very much for appreciating our work.

1-The stellate ganglion blocks and lumbar sympathetic blocks are used for diagnosis and treatment of sympathetically mediated/maintained pain in a subset of patients diagnosed with CRPS. You could mention about stellat block in the discussion section.

-Thank you for this valuable suggestion. These methods have been added to the treatment section – lines 202-203

2- In CRPS I, the pain is nociceptive, and in CRPS II, it is neuropathic. 

I do not agree with this sentence. Could you please add reference or rewrite the sentence.

-Thank you. The statement has been corrected, and a reference has been added –lines 96-96

Reviewer 2 Report

Comments and Suggestions for Authors

Abstract: 

Line 16: CRP is one of late complication of DRF

21:  perctutaneous pinning by Kirschner wire 

What is the aim of study ? you need to mention it in your abstract

introduction: 

very long ,  i suggest to start with a brief introducton then you start your case presentation , then discussion with review

99-101 : fibromyelgia and external fixation are risk factors and need to mentioned

diagnosis and epidiomiolgy : authors need to mentions bone scan as diagnostic tool 

Authors did not mention at all the rule of vitamin C !  which is a must when reviewing CRPs  and used as preventive method

these 2 study should be mentioned

 Besse, J. L., Gadeyne, S., Galand-Desmé, S., Lerat, J. L., & Moyen, B. (2009). Effect of vitamin C on prevention of complex regional pain syndrome type I in foot and ankle surgery. Foot and ankle surgery : official journal of the European Society of Foot and Ankle Surgeons15(4), 179–182. https://doi.org/10.1016/j.fas.2009.02.002

Meena, S., Sharma, P., Gangary, S. K., & Chowdhury, B. (2015). Role of vitamin C in prevention of complex regional pain syndrome after distal radius fractures: a meta-analysis. European journal of orthopaedic surgery & traumatology : orthopedie traumatologie25(4), 637–641. https://doi.org/10.1007/s00590-014-1573-2

prevention should be discussed in one paragraph anywhere in the study

photo of figures should be explained 

2.3.3 (results) :  should changed to follow-up outcome

Discussion :  first paragraph is repetition , remove it please

figure 5 , please remove exact dates as it can identify patient  ,

there is many repeatition between treatment section in introduction and discussion

case presentation :

Author Response

Abstract: 

Line 16: CRP is one of late complication of DRF

- Thank you very much for your comment. The sentence has been corrected. 

21: perctutaneous pinning by Kirschner wire 

- Thank you very much for your comment. The sentence has been corrected. 

What is the aim of study ? you need to mention it in your abstract

-The purpose of the work has been added to the abstract.

introduction: 

very long , i suggest to start with a brief introducton then you start your case presentation , then discussion with review

-Thank you for your comment. All replicates have been removed. We consider all other issues described in the introduction to be crucial in understanding the subject of our work, and further reducing the review of the literature in the introduction may negatively affect the quality of the work - as our work is a combination of a mini-review of the literature and a presentation of the patient's case.

99-101 : fibromyelgia and external fixation are risk factors and need to mentioned

-Thank you, it has been added – line 154

diagnosis and epidiomiolgy : authors need to mentions bone scan as diagnostic tool 

- Thank you for this comment. Bone scans have been added – lines 148-149.

Authors did not mention at all the rule of vitamin C ! which is a must when reviewing CRPs and used as preventive method

-Thank you for this valuable comment – vitamin C has been added in the new prevention section – lines177-183

these 2 study should be mentioned

 Besse, J. L., Gadeyne, S., Galand-Desmé, S., Lerat, J. L., & Moyen, B. (2009). Effect of vitamin C on prevention of complex regional pain syndrome type I in foot and ankle surgery. Foot and ankle surgery : official journal of the European Society of Foot and Ankle Surgeons15(4), 179–182. https://doi.org/10.1016/j.fas.2009.02.002

Meena, S., Sharma, P., Gangary, S. K., & Chowdhury, B. (2015). Role of vitamin C in prevention of complex regional pain syndrome after distal radius fractures: a meta-analysis. European journal of orthopaedic surgery & traumatology : orthopedie traumatologie25(4), 637–641. https://doi.org/10.1007/s00590-014-1573-2

Thank you. Studies have been cited in the text and added to the bibliography.

 prevention should be discussed in one paragraph anywhere in the study

-Thank you. The prevention section has been added – lines 177-183

photo of figures should be explained

-Thank you; the descriptions have been corrected.

2.3.3 (results) : should changed to follow-up outcome

-Thank you. The chapter title has been changed.

Discussion : first paragraph is repetition , remove it please

-Thank you the first paragraph has been removed.

figure 5 , please remove exact dates as it can identify patient 

-Thank you; the dates have been removed.

there is many repeatition between treatment section in introduction and discussion

- Thank you, all replicates have been removed.

case presentation :

Reviewer 3 Report

Comments and Suggestions for Authors

1. This paper doesnot provide new information, as CRPS is a well known complication in DRFs.

2. The presentation of the paper is better and the review of the literture part has some significance that reader will know how to approach a patient witg CRPS and classify these issue.

3. The case report doesnot hold any novelty. And lack strong argument on why to publish that case repor. Why does it differ from other published case reports. Pertinent difference on the case presentation needs to be highlighted inorder to accept a case report for publication.

4. This paper has a very powerful literture review section. But has weak case report section.

R5. References arw uptodate and relevant 

6. No issue with figures and tables.

Author Response

  1. This paper doesnot provide new information, as CRPS is a well known complication in DRFs.

Thank you for this comment. Our work aimed to apply currently available treatment methods in medical practice.

  1. The presentation of the paper is better and the review of the literture part has some significance that reader will know how to approach a patient witg CRPS and classify these issue.

Thank you very much for your comment.

  1. The case report doesnot hold any novelty. And lack strong argument on why to publish that case repor. Why does it differ from other published case reports. Pertinent difference on the case presentation needs to be highlighted inorder to accept a case report for publication.

Thank you for your comment. However, we disagree with this opinion. It should be noted that one study in the available literature assesses the impact of BEMER on the treatment of CRPS. Most of them are review papers. Our case report is the first work of this type to describe the use of this method in medical practice. Our dissuasion and conclusions have been supplemented with this information. 

  1. This paper has a very powerful literture review section. But has weak case report section.

Thank you for this valuable comment – the whole case study section has been rewritten.

5. References arw uptodate and relevant 

Thank you very much for your comment.

  1. No issue with figures and tables.

Thank you very much for your comment.

Round 2

Reviewer 3 Report

Comments and Suggestions for Authors

The response is Appreciated